# The Role Played by Autophagy in FcεRI-Dependent Activation of Mast Cells

**DOI:** 10.3390/cells13080690

**Published:** 2024-04-16

**Authors:** Anastasia N. Pavlyuchenkova, Maxim S. Smirnov, Boris V. Chernyak, Maria A. Chelombitko

**Affiliations:** 1Belozersky Institute of Physicochemical Biology, Moscow State University, Moscow 119992, Russia; anabella.gen@gmail.com (A.N.P.);; 2Faculty of Bioengineering and Bioinformatics, Moscow State University, Moscow 119992, Russia

**Keywords:** mast cells, autophagy, FcεRI-dependent activation, mitochondria

## Abstract

The significant role of mast cells in the development of allergic and inflammatory diseases is well-established. Among the various mechanisms of mast cell activation, the interaction of antigens/allergens with IgE and the subsequent binding of this complex to the high-affinity IgE receptor FcεRI stand out as the most studied and fundamental pathways. This activation process leads to the rapid exocytosis of granules containing preformed mediators, followed by the production of newly synthesized mediators, including a diverse array of cytokines, chemokines, arachidonic acid metabolites, and more. While conventional approaches to allergy control primarily focus on allergen avoidance and the use of antihistamines (despite their associated side effects), there is increasing interest in exploring novel methods to modulate mast cell activity in modern medicine. Recent evidence suggests a role for autophagy in mast cell activation, offering potential avenues for utilizing low-molecular-weight autophagy regulators in the treatment of allergic diseases. More specifically, mitochondria, which play an important role in the regulation of autophagy as well as mast cell activation, emerge as promising targets for drug development. This review examines the existing literature regarding the involvement of the molecular machinery associated with autophagy in FcεRI-dependent mast cell activation.

## 1. Introduction

Mast cells (MCs) constitute an important population of connective tissue cells involved in maintaining homeostasis and contributing to both innate and adaptive immune responses [1]. Their pivotal role in the pathogenesis of allergic and inflammatory diseases is widely recognized [2,3]. The primary and extensively studied mechanism of MC activation involves the interaction of antigens/allergens with IgE, leading to its subsequent binding to the high-affinity receptor FcεRI. This triggers intracellular FcεRI signaling, culminating in prompt exocytosis of granules packed with preformed mediators, including biogenic amines (histamine, serotonin, and, in some animals, dopamine), glycosaminoglycans (heparin, chondroitin sulfate, and hyaluronic acids), and an array of enzymes (proteolytic enzymes, oxidases, decarboxylases, and phosphatases) [4]. Subsequent to FcεRI-dependent MC activation is the synthesis and release of newly formed mediators, comprising lipid mediators and a diverse repertoire of cytokines (e.g., IL-4, IL-5, IL-9, IL-13, TSLP, and TNF-α), chemokines (e.g., CCL-3, CCL-5, and CX3CL-1), and growth factors (e.g., TGF-β1, SCF, and VEGF) [5,6,7]. Notable among the lipid mediators produced by MCs are leukotriene C4 (LTC4), prostaglandin D2 (PGD2), and platelet-activating factors, which are recognized for their significant roles [8].

The binding of high-affinity IgE receptor (FcεRI) with the complex of antigens/allergens triggers the activation of proximal FcεRI-associated Src kinases (Lyn or Fyn) and the central hub kinase Syk. This leads to the phosphorylation and activation of phospholipase C (PLC)γ and phosphoinositide 3-kinase (PI3K). PLCγ activation induces Ca^2+^ mobilization and subsequently activates protein kinase C (PKC), which, in turn, mediates degranulation and the production of lipid mediators. On the other hand, PI3K converts phosphatidylinositol-4,5-bisphosphate to phosphatidylinositol-3,4,5-triphosphate, leading to the activation of various signaling kinases like Bruton’s tyrosine kinase (BTK), 3-phosphoinositide-dependent protein kinase (PDK)1, and Akt. These signaling pathways lead to the activation of downstream MAP kinases (MAPK) and mammalian targets of rapamycin (mTOR), resulting in the production of lipid mediators and cytokines. Figure 1 shows the main stages of FcεRI-dependent activation of MCs [7,9,10].

Recent findings highlight the growing evidence for the involvement of autophagy in MC activation. Autophagy is a fundamental cellular process responsible for the destruction of damaged organelles and macromolecules within lysosomes. This occurs through the formation of a double-membrane organelle called an autophagosome, which engulfs the target portion of the cytoplasm [9]. Traditionally viewed as a degradation pathway, new evidence suggests that the molecular machinery of autophagy also plays a role in cellular secretion. Thus, overexpression or inhibition of key autophagy proteins alters the secretory profile of cells [11,12].

Recently, the underlying mechanisms and regulation of autophagy have been comprehensively reviewed [11,12,13]. Briefly, autophagy is activated by cellular stress and nutrient deficiency through signaling via the AMP-activated protein kinase (AMPK) and the mammalian target of rapamycin complex 1 (mTORC1). The inhibition of mTORC1 leads to the activation of the initiation Unc-51-like kinase 1 (ULK1) complex, including the kinases ULK1/2 and the autophagy-related gene (ATG) 13, and the protein ATG101 and FIP200 ULK1 complexes are involved in the assembly of the nucleation complex (III class PI3K complex), consisting of Beclin-1, ATG14 proteins, PI3Ks Vps34, and Vps15. Phosphatidylinositol-3-phosphate produced in this complex promotes nucleation of the phagophore membrane and its ATG9-dependent elongation, as well as the recruiting of ubiquitin-like complex ATG12/ATG5/ATG16L1. ATG12 is activated by ATG7, an E1-like ubiquitin-activating enzyme. In the next step, ATG12 conjugates to ATG5 through its activation by ATG10 (E2 enzyme). ATG16L1 interacts with ATG12/ATG5. The resulting ATG12/ATG5/ATG16L1 complex participates in phagophore elongation, and after completion of its formation, the components of the complex separate and return to the cytoplasm. A second ubiquitin-like complex is formed by modification of the LC3 protein. Cytoplasmic LC3-I binds to phosphatidylethanolamine via ATG7 and ATG3, followed by cleavage of an amino acid residue and conversion to the lipidated form LC3-II. The lipidated LC3-II is selectively incorporated into the forming autophagosomal membrane and remains associated with the autophagosome until its fusion with the lysosome when LC3-II bound to the outer membrane dissociates with the help of ATG4 and recycles, but LC3-II remains associated with the inner membrane and degraded by lysosomal proteases together with the autophagosomal cargo. This is why the LC3-II is a widely used marker for autophagosome detection. Another popular marker is the autophagy receptor protein p62, which recognizes aggregates of polyubiquitinated proteins and binds to LC3-II, allowing the phagophore to engulf these aggregates and form an autophagosome. Since the p62 protein is degraded by autophagy and accumulates when autophagy is inhibited, a decrease in p62 levels indicates the successful completion of autophagy [12]. In the final stages, the autophagosome may fuse with the lysosome for degradation by lytic enzymes or be transported to the plasma membrane for further secretion. The difference in the final fate of the autophagosome relies primarily on the functioning of the protein syntaxin-17 (STX17), vesicle-trafficking protein SEC22B, and small GTPases Rab37 and Rab27 [14,15,16]. STX17 promotes the fusion of the autophagosome with the lysosome in the degradative pathway, whereas SEC22B, Rab37, and Rab27 facilitate the fusion of the secretory autophagosome with the plasma membrane. The main steps of autophagy are summarized in Figure 2.

Secretory autophagy is one of the ways of unconventional protein secretion. MC granules are secretory lysosomes in origin, and their secretion mechanism may use at least some components of the autophagic machinery [12].

A compartment for unconventional protein secretion (CUPS) is formed by endosomal as well as autophagic components, such as LC3-II and ATG9. CUPS formation is induced by nutrient starvation but not by rapamycin, which induces autophagy [17]. CUPS-dependent secretion is not involved in autophagic degradation but can contribute to the release of secretory granules from different cell types. For example, the secretion of granules containing antimicrobial proteins, including lysozyme from intestinal Paneth cells [18]; the secretion of Weibel–Palade bodies containing von Willebrand factor from endothelial cells [19]; and the secretion of lysosomes of osteoclasts [20] critically depend on autophagy.

Importantly, active autophagic degradation inevitably causes the depletion of several components involved in both autophagy and granule secretion; for example, p62 and LC3-II [21,22,23]. This phenomenon underlies the complex role of autophagy in MC activation. In this review, we consider data indicating the role of autophagy and the proteins involved in its implementation in the FcεRI-dependent activation of MCs both at the stage of degranulation and at the stage of the formation of newly synthesized mediators.

## 2. Evidence for the Involvement of Autophagy in MC Activation

The first evidence for the involvement of autophagy in MC physiology was presented in 2011 [22]. Using MCs derived from the bone marrow (BM) of transgenic mice expressing GFP-LC3, the authors demonstrated that LC3-II colocalizes with secretory granules. In addition, using MCs isolated from mouse BM and peritoneal cavity, as well as human MCs of the LAD2 line, it was shown that the conversion of LC3-I to LC3-II, a key step in autophagosome formation, occurs constitutively in MCs [22]. Since key components of the autophagy mechanism, in particular, LC3, are involved in both degradative autophagy and secretory autophagy, the question of which autophagy is observed during MC activation requires clarification.

Transcription factors such as STAT5, GATA1/2, FOG-1, and microphthalmia-associated transcription factor (MITF) play important roles in the differentiation of MCs and the maturation of their granules. MITF plays a special role in MC activation by controlling the expression of genes encoding many preformed mediators, including chymases (mMCP1, mMCP2, mMCP4, and mMCP5), tryptase (mMCP6), granzyme B, and cathepsin G [24]. In addition, MITF regulates the expression of the enzyme histidine decarboxylase, which is necessary for the production of histamine, one of the key MC mediators [25]. There is evidence that MITF expression is controlled by proteins involved in autophagy. Thus, LC3-II, ATG7, and Beclin-1 are positively associated with MITF expression and its transcriptional activity [26]. It can be assumed that autophagy-related proteins control the activity of MITF in MCs, thereby affecting the formation of preformed mediators.

Importantly, the activation of AMPK, a key inducer of autophagy, has been shown to interfere with FcεRI-dependent MC activation [27,28]. AMPK activator 5-aminoimidazole-4-carboxamide-1-b-4-ribofuranoside (AICAR) attenuated the FcεRI-dependent phosphorylation of PLCg1, ERK, JNK, IKK, mTOR, and S6K but not that of Gab2, Akt, and p38, suggesting that AMPK interferes with the core of FcεRI signaling. As a result, AICAR markedly reduced FcεRI-mediated degranulation, the generation of LTC4 and PGD2, as well as TNF-a and IL-6 secretion. On the other hand, a deficiency of AMPK isoform AMPKa2 leads to increased IgE/antigen-dependent activation of MCs [27]. Inhibition of AMPK accompanies antigen-dependent stimulation of MCs through the binding of orphan nuclear receptor 4A1 (NR4A1) with the LKB1/AMPK complex. Concomitantly, NR4A1 can interact with Syk and stimulate Syk-dependent MC activation [28]. Substances that activate the LKB1/AMPK signaling pathway reduce both the degranulation and the secretion of cytokines and lipid mediators by MCs [27,29]. The important AMPK target mTORC1, which is an inhibitor of autophagy, is activated during FcεRI-dependent stimulation in a PI3K-Akt-Erk1/2-dependent manner [9]. The inactivation of AMPK following antigen stimulation also leads to the activation of mTORC1.

Overall, these data suggest that the autophagic degradative flux must be suppressed, while several components of the autophagic machinery are required for antigen-dependent MC activation.

Two approaches are widely used to study the role of autophagy in various processes: knockout animals [30] and small-molecule autophagy inhibitors or activators [31]. Since most mice with knockout key genes involved in autophagy die before birth or immediately after, other approaches are used to create knockout models.

Evidence of the involvement of autophagy in FcεRI-dependent MC activation based on the use of small-molecule autophagy regulators should be treated with some caution. Thus, class I and class III PI3K inhibitors (wortmannin, LY294002, and 3-methyladenine) are widely used as autophagy inhibitors, but class I PI3K plays a critical role in antigen-dependent MC activation [32,33]. Less is known about the effects of ULK1/2 kinase inhibitors (MRT68921, SBI-0206965, ULK-101) on MC physiology. mTORC1 inhibitors (rapamycin, Torin-1, and AZD8055) activate autophagy, and their effects on MC activation will be discussed below. Among AMPK activators that stimulate autophagy, metformin is the most studied, but it does not have high specificity and has a number of important targets [31,34].

### 2.1. Inhibition of Autophagy Promotes MC Activation

One of the key studies examining the role of autophagy in FcεRI-dependent MC activation, already cited above [22], was conducted in mice with IFN-induced deletion of floxed ATG7. MCs obtained from the BM of these mice have impaired antigen-dependent degranulation, while the secretory granules and antigen-dependent production of newly formed mediators, such as LTC4 and cytokines TNF and IL-6, are not affected. ATG7 deletion had no effect on calcium mobilization or other early events in the FcεRI-dependent signaling cascade but prevented the conversion of LC3-I into LC3-II, while in the control cells, the LC3-II colocalized with secretory granules, and in ATG-deleted cells, LC3-II formed large aggregates diffusely distributed throughout the cytoplasm. These results suggest that autophagic machinery is involved specifically in the process of FcεRI-dependent degranulation, but not granule maturation or the production of newly formed mediators [22]. It is possible that LC3-II is required for granule exocytosis. There is also evidence that the Rab27a GTPase involved in secretory autophagy [16] is colocalized with MC granules and is required for their secretion [35,36].

Additional evidence for the involvement of autophagy-related proteins in the process of FcεRI-dependent MC activation was obtained using miRNA (miR) mimic technology [23]. An increase in the level of autophagy-related proteins (ATG5, LC3-II, pBeclin-1Ser14, and p62) was observed upon antigenic stimulation of rat basophilic leukemia cell RBL-2H3 cells, which is widely used as a model of FcεRI-dependent MC and basophil activation, as well as lung MCs whereby the absence of a decrease in the p62 suggests that degradation autophagy is reduced. Notably, p62 colocalizes with LC3-II on the secretory granules of RBL-2H3 cells. miR-135-5p was first shown to suppress p62 expression by binding to the 3’-UTR of the p62 gene. The miR-135-5p mimic was then shown to inhibit p62 expression and prevent antigen-dependent autophagic flux and signs of allergic inflammation in a mouse model of passive cutaneous anaphylaxis. In addition, these data were confirmed in RBL-2H3 cells [23]. As a precaution, however, it should be noted that miR-135-5p suppresses various genes such as SMAD3 (a mediator of TGF-β signaling) [37] and hypoxia-inducible factor 1 Inhibitor α (HIF1AN) [38].

Rather, all these data point to the suppression of degradative autophagy and the involvement of autophagic proteins in the unconventional secretion of granules during MC activation. This is consistent with data on AMPK inactivation and mTORC1 activation during antigen-dependent stimulation of MCs. The suppression of autophagy during MC activation possibly allows the autophagy-related proteins to participate in the secretion of MC granules.

Autophagy inhibitors that activate mTORC1 and inhibit mTORC2, such as MHY1485 and 3-benzyl-5-((2-nitrophenoxy) methyl)-dihydrofuran-2(3H)-one (3BDO), reduce antigen-dependent degranulation of murine BM-derived MCs (BMMCs), as well as the expression and secretion of TNF and IL-6 [39,40]. In contrast, the constitutive activation of mTORC1 and inactivation of mTORC2 by deletion of tuberous sclerosis 1 (TSC1) have been shown to reduce FcεRI-dependent degranulation but increase TNF and IL-6 secretion in BMMCs [41]. The different effects of MHY1485 and 3BDO compared to TSC1 deficiency on cytokine production suggests that the TSC1/TSC2 complex may be involved in other regulatory pathways in addition to mTORC1.

It is interesting to note that there is evidence for the participation of the mTORC1-mediated signaling pathway in the process of MC granule biogenesis after degranulation [42]. Presumably, this effect is mediated by mTORC1-dependent activation of the transcription factor EB (TFEB), which regulates autophagy and lysosome biogenesis. This transcription factor is critical for secretory granule biogenesis in MCs [43]. The inhibition of mTORC1 leads to the translocation of TFEB to the nucleus, resulting in the expression of genes involved in secretory granule biogenesis [25]. It was shown that a decrease in mTORC1 activity in MCs from the BM of mice deficient in the lysosomal amino acid/oligopeptide transporter SLC15A4 leads to increased expression and nuclear translocation of TFEB and more potent FcεRI-dependent degranulation of secretory granules [44]. Apparently, after completion of antigen-induced degranulation, mTORC1 is inactivated, leading to TFEB translocation and the formation of new secretory granules. What causes mTORC1 inhibition remains unknown. mTORC1 inhibition and TFEB activation stimulate autophagy, which, in turn, stimulate the activity of transcription factor MITF, involved in the expression of pre-formed mediators [24,25,26]. Thus, it is possible that the activation of autophagy is necessary for the recovery of secretory granules after MC activation.

In both human and rodent BMMCs, PI3K class I and III inhibitors wortmannin and LY294002, which inhibit autophagy, have been shown to reduce FcεRI-dependent degranulation and secretion of IL-6, IL-13, and TNF [9,33,45]. However, the results obtained using PI3K inhibitors should be evaluated critically since their effect may be due to the suppression of class I PI3K activity, which is critical for FcεRI signaling [32]. Thus, PI3K-dependent activation of Akt, Btk, PLD, and Ca^2+^-mobilization are required for both degranulation and cytokine production [32,33].

At the same time, another PI3K class III inhibitor 3-methyladenine, which is an autophagy inhibitor, enhanced degranulation and IL-4 and IL-6 cytokine production in LAD2 MCs, stimulated by compound 48/80 [46].

No data are available on the effects of ULK1/2 inhibitors on antigen-dependent MC activation. However, it has been shown that the ULK1/2 inhibitor MRT68921 reduces degranulation of MC lines RBL-2H3 and LAD2 induced by G protein-coupled receptor MRGPRX2 ligands, which activate MCs. MRT68921 also prevents Erk1/2 phosphorylation during MRGPRX2-induced degranulation, indicating a role for ULK1 in Erk1/2 phosphorylation [47]. These data support an important role for ULK1/2 in the degranulation process. The inhibitory effect of MRT68921 on MC degranulation may be related both to the suppression of Erk1/2 activity, which is required for degranulation, and to non-canonical functions of ULK1/2 unrelated to autophagy. For example, ULK1/2 is known to be involved in endoplasmic reticulum-to-Golgi vesicular transport and some other intracellular vesicular trafficking [48].

Thus, MC granules contain markers of autophagosomes (LC3-II and p62), as well as parts of markers responsible for autophagosome secretion, such as Rab27a and Rab37a. Whether there is another important autophagosome secretory marker Sec22b in MC granules is not known. AMPK activity inhibition and mTORC1 stimulation, which are positive and negative regulators of autophagy, are observed during MC activation; autophagy suppression should be observed. The absence of p62 degradation during MC activation is consistent with this assumption. This indicates the involvement of autophagy-related proteins in MC granule secretion.

At the same time, autophagy plays a much less unambiguous role in the secretion of newly formed MC mediators than degranulation. It has been shown [22] that deletion of the key autophagy gene ATG7 in MC does not impair the secretion of newly formed mediators, while the degranulation process is suppressed. It should be noted that some cytokines, such as bFGF, IL-4, SCF, and TNF, can accumulate in secretory granules simultaneously with their formation after FcεRI-dependent activation [8]. Importantly, the production of cytokines, chemokines, and growth factors in MC is dependent on the activity of transcription factors such as NF-kB, NFAT1/2, PU.1, GATA1/2, Ikaros (zinc finger transcription factor), BATF, and STAT5 [5,6,7], which can be regulated by autophagy. For example, the transcription factor PU.1, which stimulates IL-4 expression in MCs [49], is degraded during autophagy [50]. The adaptor protein Bcl-10, involved in the activation of the transcription factors NFκB and AP-1 in MCs [51], can also undergo degradation via the autophagic pathway [52]. This indicates that the expression of IL-4, TNF, IL-6, etc., should be increased when autophagy is inhibited. Indeed, the autophagy inhibitor 3-methyladenine has been shown to enhance IL-4 and IL-6 production induced by compound 48/80 [46]. Antigen-dependent secretion of TNF and IL-6 increases in Tsc1-deficient MCs with inhibited autophagy [41].

The secretion of cellular proteins, including cytokines, typically requires an N-terminal signal peptide for delivery to the endoplasmic reticulum and subsequent secretion. However, some proteins, in particular IL-1β family cytokines, do not have a signal peptide and are secreted through secretory autophagy [53]. It can be expected that the secretion of IL-1β and some other cytokines in MCs also depends on secretory autophagy. If this is the case, then the inhibition of autophagy during MC activation would inhibit the secretion of these proinflammatory cytokines. This mechanism may serve as protection against excessive inflammation during allergic activation of MCs.

### 2.2. Autophagy Activation Interferes with MC Activation

The data discussed above indicate that the activation of AMPK by AICAR, which stimulates autophagy, interferes with antigen-dependent MC degranulation and cytokine secretion [27]. Another AMPK activator, metformin, also reduces the antigen-dependent secretion of IL-13 and TNF-α in mouse BMMC and suppresses the passive cutaneous anaphylaxis reaction in mice [54]. In addition, metformin reduced the manifestations of ovalbumin-induced asthma in mice, including the levels of IL-4, IL-5, IL-13, TNF-α, and TGF-β in bronchoalveolar lavage [55]. However, these data should be treated with caution because AMPK has many targets other than autophagy and its activators are highly non-specific.

It has been reported that the widely used mTORC1 inhibitors rapamycin and Torin 1, which stimulate autophagy, do not affect antigen-dependent MC degranulation [9,56]. It is likely that MC degranulation depends on CUPS-dependent secretion, which is known to promote granule release in other cell types and is not affected by Rapamycin [57]. At the same time, rapamycin reduces the FcεRI-dependent secretion of IL-6 and IL-8 by MCs obtained from human BM and peripheral blood [9]. These data are in contrast to studies using the TORC1 activators MHY1485 and 3BDO, where they were shown to inhibit both antigen-dependent degranulation and the secretion of TNF and IL-6 in MC [39,40]. The reason for this discrepancy likely reflects the complex network of signaling pathways controlled by mTORC1 and mTORC2 and deserves further study.

A number of studies show that the physiological modulation of autophagy influences MC activation. For example, overexpression of orosomucoid-like-3 (ORMDL3) protein, which activates the unfolded protein response and autophagy, prevents antigen-dependent activation of MC line MC/9 and significantly suppresses degranulation, as well as cytokine (IL-6, IL-13, and TNF-α) and chemokine (CCL3 and CCL4) production. Inhibition of autophagy by 3-methyladenine abolished this effect. The knockdown of ORMDL3 enhanced the passive cutaneous anaphylaxis response in mouse ears, indicating that ORMDL3 suppresses antigen-dependent MC activation in vivo [58]. It should be noted that ORMDL3 is strongly associated with asthma [59].

IL-33, an important immune modulator that stimulates inflammation in allergic disease, inhibits autophagy and promotes degranulation of MC in the model of allergic rhinitis [46]. The suppression of autophagy associated with MC activation has also been described in a model of secondary infection with Staphylococcus aureus after infection with the H1N1 influenza virus. There was an increased release of histamine and the most abundant serine proteinase of secretory granules, tryptase, and various cytokines were observed in the lungs of co-infected mice [60].

Thus, the above data suggest that modulation of autophagy may be an important physiological way to regulate MC activation. Figure 2 shows the scheme of autophagy processes associated with FcεRI-dependent activation of MCs.

Table 1 summarizes the effects of autophagy inhibitors and activators on MC activation. In Figure 2, we show the targets of the main inhibitors of autophagy pathways.

## 3. Mitochondria at the Crossroads of Autophagy and MC Activation

Mitochondria are an important hub of intracellular signaling and can significantly modulate MC activation, as revised in Ref. [63]. At the same time, mitochondria contribute to the regulation of autophagy [64,65], so it is likely that the regulation of MC function by mitochondria is mediated by autophagy. Mitochondria influence MC activation by supporting their energy metabolism and calcium signaling [66]. The transcription factors STAT3 and MITF, which are involved in MC activation, are partially localized to mitochondria, and their mitochondrial activity has been shown to influence FcεRI-dependent MC activation [67,68]. Mitochondrial fragmentation and localization changes have been observed during MC activation and are thought to be involved in the regulation of MC degranulation but do not affect de novo mediator biosynthesis [69]. It is known that MC activation depends on the redox status of the cells and the production of reactive oxygen species (ROS), the main source of which is mitochondria [70]. Our studies using the mitochondria-targeted antioxidant SkQ1 showed that mitochondrial ROS (mtROS) production is critical for MC degranulation both in vitro and in vivo [71]. Another mitochondria-targeted agent, C_12_TPP, at very low concentrations, prevents mitochondrial dysfunction, including fragmentation, and inhibits IgE-dependent degranulation of RBL-2H3 cells [72]. Since mitochondrial fragmentation can be mediated by an increase in mtROS [73], it is possible that C_12_TPP, like SkQ1, inhibits the generation of mtROS. This antioxidant effect is likely mediated by partial (mild) uncoupling caused by TPP^+^-based compounds [74].

Mitochondria are known to contribute significantly to the regulation of autophagy [64,65], so it is likely that the regulation of MC function by mitochondria is dependent on autophagy. First of all, the cessation of oxidative phosphorylation in mitochondria leads to the activation of AMPK due to an increase in the level of AMP. mtROS can also stimulate autophagy due to the inhibition of Akt/mTOR signaling [75] and AMP-independent activation of AMPK mediated by oxidation and further glutathionylation of critical cysteine residues [76,77]. Additionally, mtROS has been shown to activate AMPK through phosphorylation by the serine/threonine protein kinase LKB1 under hypoxia [78]. Moreover, mtROS can inhibit phosphatase PTEN, which interferes with autophagy through suppression of the PI3K-dependent pathway [79].

On the other hand, mtROS has been shown to stimulate Erk1/2 [72], which suppresses AMPK [80] and activates mTORC1 [81], thus inhibiting autophagy. Interestingly, mitochondrial fragmentation has been shown to depend on the activity of Erk1/2 during antigen-dependent stimulation of RBL-2H3 cells [72]. This observation presumably reflects Erk1/2-dependent inhibition of mitochondrial autophagy.

Over the last two decades, a large array of compounds has been developed that specifically target mitochondria and modulate their functions [82]. Since mitochondria play dual roles in regulating autophagy and MC activation, they become promising targets for drug development.

## 4. Conclusions

Thus, an analysis of the literature data suggests that antigen-dependent activation of MCs is accompanied by a decrease in autophagy. Inhibition of autophagic degradation allows some proteins of the autophagic machinery to participate in the secretion of MC granules and prevents the degradation of transcription factors involved in the production of cytokines. The molecular mechanisms underlying the role of autophagy in antigen-dependent MC activation remain poorly understood. Further studies using transgenic mice deficient in key autophagy genes and expressing fluorescent protein probes for autophagy may help elucidate these mechanisms. Studies using transgenic mice deficient in key autophagy genes, such as Becn1, Atg13, ULK1/2, Pik3c3/Vps34, Atg3, Atg5, Atg12, and Atg16l1, can also significantly stimulate research progress. In addition, various small molecule inhibitors and activators of autophagy, especially ULK1/2 inhibitors, are worthy of study. At the same time, some small molecule activators of autophagy demonstrate an inhibitory effect on antigen-dependent activation of MCs, as well as a therapeutic effect in a number of allergic animal models.

In this regard, further research in this area will not only expand the fundamental understanding of the mechanisms of antigen-dependent MC activation but are also of interest to medicine.

## Figures and Tables

**Figure 1 cells-13-00690-f001:**
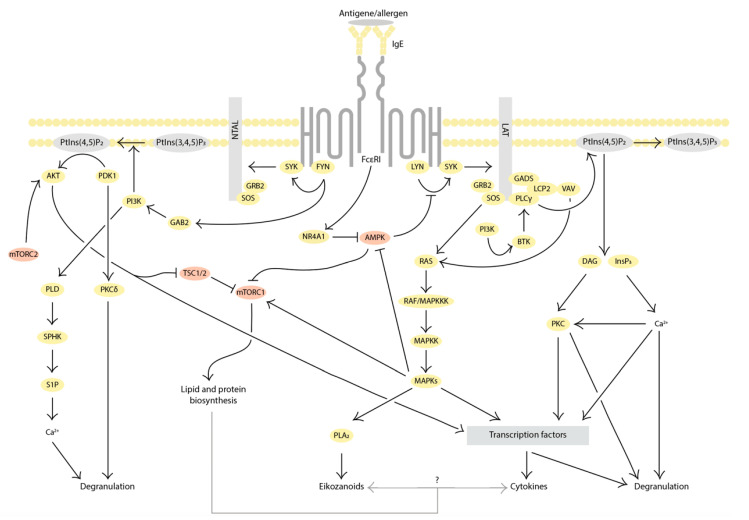
The main stages of FcεRI-dependent activation of MCs. The transmembrane adaptors LAT and NTAL are phosphorylated upon the IgE/antigen complex, binding to FcεRI. Activation occurs through the LAT-dependent pathway, while the NTAL-dependent pathway provides maintenance of the MC-activated state. Ligand-dependent aggregation of FcεRI causes the phosphorylation of tyrosine in immunoreceptor tyrosine-based activation motifs (ITAMs) in their cytoplasmic tails by the LYN kinase. The SYK kinase is then recruited to phosphorylate LAT and NTAL. Activation of LAT initiates the assembly of a signaling complex containing growth factor receptor-related protein 2 (GRB2), GRB2-associated adaptor protein (GADS), lymphocyte cytosolic protein 2 (LCP2, also known as SLP76), phospholipase Cγ (PLCγ), and the guanine nucleotide exchange factors SOS and VAV. PLCγ catalyzes the hydrolysis of phosphatidylinositol-4,5-bisphosphate (PtdIns(4,5)P2) in the plasma membrane. The inositol-1,4,5-triphosphate (InsP3) and diacylglycerol (DAG) formed in this reaction induce calcium release from intracellular depots into the cytosol and activate protein kinase C (PKC). LAT also activates the small GTPase RAS, which activates the RAF kinase, followed by MEK kinases, and, finally, mitogen-activated protein kinases (MAPKs) such as ERK1, ERK2, p38, and JNK. These, in turn, affect transcription factors such as AP-1, NFAT, and NF-κB, which trigger the production of many cytokines. Among others, RAS activates phospholipase A2 (PLA2), which regulates eicosanoid synthesis. There is also a LAT-independent pathway of MC activation carried out by the FYN kinase, which phosphorylates the adaptor molecules NTAL and GAB2, the latter of which activates phosphatidylinositol-3-kinase I class (PI3K). PI3K can induce calcium mobilization in several possible ways: through the recruitment of BTK kinase followed by PLCγ signaling or through phospholipase D (PLD), which activates sphingosine kinase (SPHK), leading to the formation of the sphingolipid mediator sphingosine-1 phosphate (S1P) from sphingosine. S1P is involved in the mobilization of calcium from intracellular depots in an InsP3-independent manner. PI3K also activates PKCδ, which regulates MC degranulation, and PDK1 kinase, which stimulates Akt kinase. Akt activates the transcription factor NF-κB, which is responsible for the expression of various cytokines, and inhibits the TSC1/2 complex, which leads to the activation of the mammalian target of rapamycin complex 1 (mTORC1) that enables protein and lipid synthesis. Importantly, FcεRI-dependent MC activation depends on the inactivation of AMP-activated protein kinase (AMPK) via orphan nuclear receptor 4A1 (NR4A1). The inactivation of AMPK following antigen stimulation also leads to activation of mTORC1. Proteins involved in autophagy-related signaling and FcεRI-signaling are marked in pink [7,9,10].

**Figure 2 cells-13-00690-f002:**
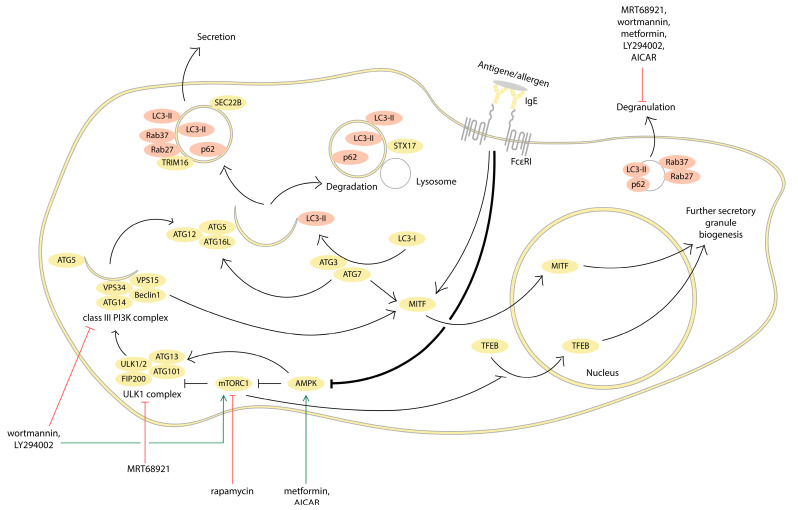
The scheme of autophagy processes associated with FcεRI-dependent activation of MCs. Activation of FcεRI-dependent signaling leads to inhibition of AMPK kinase, followed by activation of the mTORC1 complex and inhibition of autophagy. When mTORC1 is inactive, autophagy is initiated by the activity of the ULK1 complex, which leads to the assembly of a nucleation complex (class III PI3K complex). The class III PI3K complex and ATG9 are involved in phagophore membrane nucleation and elongation and recruit the ATG12/ATG5/ATG16L1 ubiquitin-like complex. ATG7 and ATG3 convert cytoplasmic LC3-I to the lipidized form of LC3-II, which is incorporated into forming the autophagosomal membrane. The autophagy receptor protein p62 binds to LC3-II, allowing autophagosome formation to be completed. The difference in the final fate (degradation vs. secretion) of autophagosomes depends primarily on the function of the STX17 and SEC22B proteins, as well as the small GTPases, Rab37 and Rab27 [12,14,15,16]. Some autophagy-related proteins (marked in pink) colocalize with MC secretory granules and appear to be involved in their secretion. Inhibition of autophagy during MC activation allows these proteins to participate in the degranulation process instead of degradation. After MC degranulation, mTORC1 is inhibited. This turns on autophagy and activates the transcription factors MITF and TFEB, which are involved in the biogenesis of new secretory granules during the recovery after MC activation.

**Table 1 cells-13-00690-t001:** Effect of low-molecular-weight autophagy regulators on MC activation.

Substance	Target	Cell Line	Effect on the Degranulation	Effect on the Cytokines
**Autophagy inhibitors**				
Wortmannin (30–100 nM)	PI3K (inhibition)	RBL-2H3		Suppression of IL-4 transcription and secretion [61]
Wortmannin (100 nM)		BMMCs	Suppression [9,45]	Suppression of IL-6 secretion [9,45], IL-13, and TNF secretion [45]
Wortmannin (500 nM)		RBL-2H3		No effect on the TNF-α transcription level [62]
LY294002 (25 nM)				
LY294002 (5 μM)		BMMCs	Suppression [45]	Suppression of IL-6, IL-13, and TNF secretion [45]
LY294002 (3–30 μM)		RBL-2H3		Suppression of IL-4 transcription and secretion [61]
3-methyladenine (5 mM)		LAD2	Stimulation [46]	Stimulation of IL-4 and IL-6 cytokine production [46]
MHY148 (2 μM)	mTORC1 (activation); mTORC2 (inhibition)	BMMCs	Suppression [39]	Suppression of TNF and IL-6 transcription and expression [39]
3BDO (50 μM)		BMMCs	Suppression [40]	Suppression of TNF and IL-6 transcription and expression [40]
MRT68921 (1 μM)	ULK1/2 (inhibition)	RBL-2H3 and LAD2	Suppression [47]	
**Autophagy activators**				
Rapamycin (10 nM)	mTORC1 (inhibition)	RBL-2H3		Destabilization of *TNF-α* mRNA [62]
Rapamycin (100 nM)		BMMCs (IL-6) and MCs from peripheral blood (IL-8)	No effect [9]	Suppression of IL-6 and IL-8 secretion [9]
Metformin (1–10 μM)	AMPK (activation)	BMMCs	Suppression [29]	Suppression of IL-13 and TNF-α secretion [29]
AICAR (1 mM)		BMMCs	Suppression [27]	Suppression of LTC4, PGD2, TNF-a, and IL-6 secretion [27]

## Data Availability

Not applicable.

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
