# Peer review of "The Role Played by Autophagy in FcεRI-Dependent Activation of Mast Cells"

_cells, 2024, doi:10.3390/cells13080690_

Round 1

Reviewer 1 Report

Comments and Suggestions for Authors

The review article by Pavlyuchenkova et al., provides a comprehensive summary of the literature on the role of autophagy in mast cells. This is an important and timely topic, as it is clear that alongside the role of autophagy in degradation of cytosolic aggregates or damaged organelles, this process is involved in secretory processes. There is therefore room for such a review that presents to the reader the available literature on the role of autophagy in mast cells. However, there are a number of concerns that I recommend that the authors will address.

1)    While the authors describe the literature reports, I encourage them also to discuss the data.  For example, if degranulation is inhibited, is it a general phenomenon or is it restricted to IgE-antigen -induced degranulation and if so is it related to the secretory process or to receptor signaling?

2)    The review describes the results of different papers, but does not attempt to put them together-I would recommend to summarize together, and not back and forth, all the literature reports that have looked into the impact of inhibition of autophagy by either genetic or pharmacological tools, and their conclusion(s)  versus all literature reports that describe the impact of activation of autophagy, what are their conclusions and then finally, combine everything together to discuss the conclusions.

3)    There is also room to provide the personal perspectives of the authors-what do they think on the involvement of autophagy in mediating mast cell functions.

4)    It is also important to emphasize the difference between secretory autophagy that mediates unconventional secretion of IL1b (line 121) and the role of autophagy in mast cell degranulation that occurs via regulated exocytosis of secretory granules.

5)    Figure 1 describes the signaling of the IgE receptor, which includes many details that are not directly relevant to the topic of the review, while it misses the signaling pathways, such as mTORC1 and mTORC2 and Tsc1 that are relevant to the review. I propose to replace figure 1 with a figure that clearly outlines the signaling pathways that are discussed in the review.

Minor points:

1)    Line 26: mast cells are not only found in connective tissues, but also in mucosal tissues.

2)    Line 214: wortmannin and LY294002 are inhibitors that may affect autophagy, but they are not autophagy inhibitors.

3)    A legend should be added to Figure 2 to explain the scheme-according to the scheme it appears that the role of autophagy is to induce MITF that is required for mediator synthesis. If this is the overall conclusion then the authors should say that.

4)    Lines 246-251 are an exact repetition of lines 113-118

Comments on the Quality of English Language

English is fine

Author Response

Dear Reviewer,

Thank you for your attentive attitude to our work and valuable comments.

We have changed the structure of the review in accordance with your recommendations and added a number of new data and conclusions (highlighted in yellow). This allowed us to more clearly formulate the hypothesis about the role of autophagy in the activation of mast cells.

  1. As per your suggestion, we have discussed additional data regarding the influence of autophagy on mast cell activation. These data suggest that proteins involved in autophagy play an important role in the general mechanism of mast cell granule secretion. Unfortunately, there is no data on the role of autophagy in the activation of mast cells by various ligands, so we have focused primarily on the discussion of IgE-antigen-dependent activation. Another important ligand that activates MCs is IL-33. Its signaling is distinct from FceR1-dependent but is also dependent on autophagy inhibition. These data are discussed in the revised version.
  2. In accordance with your recommendation, we have summarized extended data on the effects of autophagy inhibitors and activators. The revise version presents these data in two new sections 2.1. “Inhibition of autophagy promotes MC activation” and 2.2. “Autophagy activation interferes with MC activation”, and the conclusions are combined for discussion.
  3. We made bolder proposals about the involvement of autophagy in mast cell activation. They are based on data indicating the participation of proteins of the autophagic machinery in the secretion of mast cell granules. In this case, activation of autophagy with subsequent degradation of these proteins should interfere with mast cell activation. This is probably why antigen-dependent activation of mast cells is accompanied by inactivation of AMPK kinase and activation of the mTORC1 protein complex, which leads to inhibition of autophagy. Also, inhibition of autophagy prevents the degradation of a number of transcription factors involved in the secretion of certain cytokines. Following degranulation, mTORC1 is inactivated to stimulate autophagy and activate the transcription factors TFEB and MITF, which are involved in the biogenesis of new granules.
  4. We have only briefly discussed the secretory autophagy since there are no clear data indicating its role in release of the secretory granules. More data indicate that some proteins involved in both secretory and degradative autophagy can be recruited to the secretory granules and participate in their release. Secretory autophagy mediates secretion of IL-1b and the relative cytokines but the data on its role in secretion of main cytokines characteristic for activated mast cell are absent. There are some indications (not solid) that secretory autophagy is not strictly dependent on AMPK and mTORC1. If so, MC activation probably is not accompanied with inhibition of secretory autophagy and it can contribute to secretion of some MC mediators.
  5. We have added to Figure 1 the signaling pathways involving Tsc1 mTORC1 and mTORC2. We also added an important NR4A1-dependent pathway leading to AMPK inhibition. The signaling pathways initiated by the IgE receptor and involved in the activation of mast cells are left for the sake of completeness.

In the revised version, we took into account all the minor comments. In particular, a legend has been added to Figure 2.

We are grateful for your interest in our work and for your valuable comments. We hope that our review has now become more complete and understandable.

Reviewer 2 Report

Comments and Suggestions for Authors

This is a short but interesting review by A. N. Pavlyuchenkova et al.

Comments:

.- Despite FIGURE 1, FIGURE 2 lacks a legend; please include it explaining that it shows the targets of the main (small molecule) inhibitors/activators of autophagy pathways.

.- Please, include “metformin” in figure 2

.- There is no need to explain what an inducible knockout mouse model is (lines 143 -146). The explanation of the sentence “One such approach…. by phage recombinase” (lines 143-147) is of no interest to the potential readers of this paper, please delete it.

.- The sentences in lines 246 to 251 should be included in section 2.1 and deleted from section 3

.- Please, expand and explain in section 2.1 the sentences indicated in lines 319-323 in the conclusions.

.- Section 4 “Cytokines” is too short and needs more explanation, please expand it. For example, clarify and expand the sentence “it has been shown that autophagy can stimulate the secretion of cytokines IFN-γ, TNF-α, IL-1β and inhibit TNF-α, IL-17, IL-1β, IL-α secretion(lines 291-293) with the original citations (not only revision 58) to better explain the contradiction about “simultaneous” stimulation and inhibition of TNF-α and IL-1β. In addition, please expand on the explanation of cytokines, chemokines and growth factors production given in lines 297-300.

Other (minor) comments:

.- Please, cite the complete noun the first time an abbreviation is used.  

.- You must use a single term to define mast cells. On line 26 you define MC as an abbreviation of mast cell/s; but later on, you still use mast cell (e.g. lines 68, 75, 77, 112, 326) or mastocytes (i.e. line 42). Please, check the whole text to correct this.

.- Bone marrow (BM) abbreviation is cited twice (lines 114 and 247)

.- Please, include the full explanation for mtROS (lines 268 and 269), i.e. mitochondrial ROS, and use it also in lines 264 and 280.

.- Citation 40 (line 253) should be deleted or, instead, cited as “revised in 40”.

Author Response

Dear reviewer,

Thank you for the time you took to familiarize yourself with our work and for its positive assessment. We have tried to correct the shortcomings you have found and take into account all comments. We have significantly expanded the review (new data and their discussion are highlighted in yellow), while changing its structure.

- We have added to Figure 1 proteins involved not only in FcεRI signaling, but also related key components of the autophagy triggering and regulation pathway. We have also added the legend toFigure 2 according to your recommendations.

- We have added to Figure 2 "metformin" and indicated its targets. We also added information about AICAR, another AMPK activator, to the Fig. 2 and text.

- We have removed the details about the knockout model.

- We have moved that fragment to a more appropriate section (lines 148-153).

- We have changed the conclusion, because in the process of analyzing new data, a clearer concept was formulated.

- Due to insufficient data on the role of autophagy in cytokine production in mast cells, we decided to change the structure of the review. We deleted a separate chapter on cytokines and moved these data to sections 2.1 and 2.2.

All minor comments were also taken into account.

Thank you again for your valuable comments. We hope this made our review more understandable and useful for readers.

Round 2

Reviewer 2 Report

Comments and Suggestions for Authors

Thank you very much for considering my comments. I think this review can be very useful for readers. Congratulations on your excellent work.